# The DLC Coating on 316L Stainless Steel Stochastic Voronoi Tessellation Structures Obtained by Binder Jetting Additive Manufacturing for Potential Biomedical Applications

Dorota Laskowska [1,*] , Błażej Bałasz [1], Witold Kaczorowski [2] , Jacek Grabarczyk [2], Lucie Svobodova [3] , Tomasz Szatkiewicz [1] and Katarzyna Mitura [1,3,*]

1. Faculty of Mechanical Engineering, Koszalin University of Technology, Śniadeckich 2, 75-620 Koszalin, Poland
2. Faculty of Mechanical Engineering, Institute of Material Science and Engineering, Lodz University of Technology, Stefanowskiego 1/15, 94-924 Lodz, Poland
3. Faculty of Mechanical Engineering, Technical University of Liberec, 461 17 Liberec, Czech Republic
* Correspondence: dorota.laskowska@tu.koszalin.pl (D.L.); katarzyna.mitura@tu.koszalin.pl (K.M.)

**Abstract:** The DLC coating of samples produced by additive manufacturing with complex shapes is a challenge but also shows the possibility of obtaining a diffusive barrier for biomedical applications. In this study, stochastic porous structures based on Voronoi tessellation were fabricated using binder jetting technology from 316L SS powder and modified using DLC coating. The DLC coating was deposited using the RF PACVD technology. The chamber pressure was 40 Pa with a methane gas flow rate of 60 sccm. The negative bias voltage was 700 V. The deposition time was 5 min. Dimensional analysis was performed using optical microscopy. Surface morphology and topography were evaluated using SEM and confocal microscopy. Raman spectroscopy was used to examine the chemical structure of DLC coating. Finally, the HR TEM was used to evaluate the physicochemical characterization of the DLC coating. The interconnected complex spatial network of the Voronoi structure was accurately duplicated by the binder jetting technology. The obtained substrates were characterized by high roughness (Ra = 6.43 μm). Moreover, the results indicated that the conditions of the RF PACVD process allow for the deposition of the continuous and tightened DLC coating with a thickness from 30 nm to 230 nm and defined the content of $Cr_2O_3$ and $SiO_2$.

**Keywords:** additive manufacturing; barrier diffusion; binder jetting; Diamond Like Carbon; High-Resolution Transmission Electron Microscopy; Voronoi tessellation

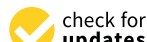



## 1. Introduction

The primary functions of the bones are to provide a mechanical support system (skeleton) for muscular activity, to provide physical protection of organs and soft tissue, to produce morphological blood components, and to ensure systematic mineral homeostasis. It is a physiologically dynamic tissue with four cell types: osteoblast, osteoclast, osteocytes, and bone lining cells. According to Wolff's law, the internal bone architecture is constantly rebuilt under the influence of load. At the structural (morphology) level, the bones of the adult person consist of cortical and cancellous (trabecular) bones. The cortical bone is characterized by low porosity (approximately 10%) and high mechanical properties in comparison to cancellous bone (50%–90% of porosity). The proportions vary in different locations in the skeleton, but the cortical bone mainly forms a layer above the cancellous bone. In the case of large bone defects after physical trauma, osteoporosis, or tumor resection, bone tissue engineering (BTE) plays an important role. The main purpose of BTE is to induce the regeneration of new bone tissue using scaffolds and growth factors. Scaffolds should meet several requirements in terms of mechanical properties, porosity, biocompatibility, degradation rate (in the case of short-term implants), osteoconduction, osteoinduction, and permeability [1–4].

One of the BTE scaffold groups is metal implants. Most of the metal bone implants and endoprosthesis currently used are made from 316L stainless steel, titanium (Ti), and its alloys or cobalt-chromium alloys (Co-Cr) because of their strength, corrosion resistance, and low immune response [5–8]. The biggest problem is a mismatch between the mechanical properties of conventional implants and natural bones, which can lead to a decrease in bone density and bone fracture or aseptic loosening of the implant. This phenomenon is referred to as stress shielding [9]. Known methods of reducing the risk of the aseptic loosening of the implants are: increasing the interfacial bond between bone tissue and implant materials, and decreasing the stiffness of the implant, via compositional or structural modification. The compositional modifications consist in introducing a β phase through doping with β stabilizers or sophisticated thermomechanical processes, while structural modifications consist in introducing the designed porosity into the material, the so-called porous structure [10,11]. The biggest problem is limited control over pore size, shape, volume fraction, and distribution. Metal powder additive manufacturing techniques such as powder bed fusion and binder jetting have overcome these limitations [11–13]. Studies have shown that the architecture of the porous implant (has a significant impact on mechanical properties and ensures the correct integration of the implant into the biological environment through the process of osseointegration and neovascularization [14–16]. The large specific surface area of the porous implant promotes cell migration and adhesion, the flow of nutrients and oxygen, and promotes a biologic bond between human bone and an implant with a porous structure [15–19]. The development of additive manufacturing (AM) significantly influences the fabrication of structures with complex or customized architecture, providing high control of internal architecture and porosity (pore size, morphology, and distribution). The wide range of available technologies ensures production from various materials, including thermoplastics, resin, ceramics, metals, and alloys [20]. Binder Jetting is AM technology in which an object is created by applying the liquid binding agent onto the selected surface area of the powder bed. The excellent packed layer of powder is laid on the build platform and the binder is applied via print head on the selected area according to the CAD model. Binder fused the powder particles and the so-called green part was created. Once the printing process is complete, the job box containing the green part and excess powder is heated to cure or solidify the binder. In addition, some components of the binder evaporate at this temperature. Then, the excess powder has to be removed in the process called "depowderization". The final stage is sintering the samples in a vacuum furnace [20–24]. An advantage of binder jetting technology is the possibility of manufacturing from different materials at room temperature and atmosphere. Moreover, elements with complex geometry can be fabricated without support. The main disadvantages are the need for post-processing to improve the mechanical properties and shrinkage during the sintering process, which can cause distortions and fractures [20,25].

A commonly used method to produce barrier diffusion on the surface of the implant is a surface modification with a Diamond Like Carbon (DLC) coating. DLC coating is amorphous carbon with σsp$^3$, σsp$^2$ and σsp$^1$ electron hybridization Properties of coating depend on a few factors: the method of synthesis, parameters of the particular method, and the material of the substrate. There are several methods of producing carbon coatings such as ion beam deposition (IBD), radio frequency plasma assisted chemical vapor deposition (RF-PCVD), filtered cathodic vacuum arc (FCVA), ion plating, plasma immersion ion implantation and deposition (PIIID), magnetron sputtering, ion beam sputtering, pulsed laser deposition and mass-selected ion beam deposition [26–30]. Numerous studies confirm the usefulness of DLC coating in biomedical applications. Due to their super tribological and mechanical properties [31–33], corrosion resistance [34], biocompatibility, and hemocompatibility [28,35]. In addition, they reduce the adhesion of pathogens and the development of biofilms on coated surfaces [36].

In the case of conventionally manufactured materials and products, DLC coating technologies are at a high level of development. Along with the progress of additive manufacturing technologies, research on the production and analysis of the properties of

DLC layers on printed substrates began. Tillmann et al. [37] analyzed the growth mechanism of DLC coating deposited onto the 316L substrates using the magnetron sputtering process. The results show that a thin DLC coating covers residual porosity (on the surface) when the pore size is smaller or equal to the coating thickness. The produced coating was characterized by high adhesion to the substrate and microhardness.

Depending on the pore size, a distinction is made between macroporosity (>50 μm) and microporosity (<20 μm). Macroporosity contributes to osteogenesis, facilitating the transport of cells and ions, while microporosity improves bone growth in scaffolds by increasing the adsorption area of proteins [3,38,39]. As shown by Woodard et al. [38] and Hing et al. [40] the best results in the regeneration of bone defects are achieved by implants that combine both types of porosity. However, most authors agree that the optimal porosity of the implant structure should exceed 50%, and the pore size should be in the range of 100 to 700 or even 1200 μm [10,12]. Most of the design of porous implants is based on regular stochastic pores [41], such as octet, body-cantered cubic (BCC), and triply periodic minimal surfaces (TPMS). The biggest advantages of this type of structures are simple design based on periodically repeated unit cells and easy control of porosity. However, they do not fully mimic the irregular architecture of bone tissue, especially trabecular bone, which is characterized by variable pore size and shape or variable porosity. Design and fabrication of irregular, porous implants bring more difficulties, especially in terms of controlling the design parameters. Nevertheless, increasing attention is being paid to the design of implants based on Voronoi tessellation- randomized pore trabecular-like porous structures.

Deering et al. [42] analyzed the histomorphometry and anisotropy of the porous scaffold on the basis of selective Voronoi tessellation. Research has shown that the Voronoi tessellation is an effective way to mimic the bone structure, by being able to determine local and global anisotropy. Moreover, it was found that the Voronoi structures imitate the natural arrangement of trabeculae in bone- similarity in terms of basic bone histomorphological indicators (porosity, trabecular thickness, thickness of the bone cortex, and surface-to-volume ratio). Chen et al. [41] investigated the permeability and stress distribution (finite element analysis) of irregular scaffolds based on Voronoi tessellation in comparison to uniform scaffolds (BCC, pillar BCC, octet). Both types of scaffolds were designed as uniform and gradient structures. The results showed that the permeability of tested scaffolds depends on their porosity and gradient irregular scaffold are characterized by the most extensive permeability coverage. This can play an important role in the transport of nutrients and oxygen to deep regions of the scaffold, which promotes better bone growth. The morphology and mechanical properties of the irregular porous structures were also analyzed by Du et al. [16] while determining the relationship between main design parameters and porosity, apparent elastic modulus, and compressive strength. The structures were manufactured by selective laser melting technology from Ti6Al4V alloy. The results showed that the design parameters that significantly influenced porosity were strun diameter, distance between seed points, and irregularity. The porosity decreases with an increase in the strun diameter and decreases with the distance between seed points. Moreover, the strun diameter has a significant influence on the apparent modulus of elasticity. Research that also deserves special attention is Liang et al. [43] work, which investigated the morphology, mechanical properties, and in vitro biocompatibility of Ti6Al4V scaffolds based on Voronoi tessellation produced with the use of selective laser melting technology. The results of in vivo examination performed on MG63 cell culture showed that irregular scaffolds with high porosity exhibited enhanced proliferation and differentiation for osteoblast cells, thanks to the combination of small and large pores with various shapes. All authors emphasize the usefulness of Voronoi structures in bone tissue engineering.

In the process of osseointegration, not only mechanical properties and a porous structure are important. Surface properties such as roughness also play an important role. Research shows that high surface roughness promotes the adhesion of cells and proteins [44–46]. Depending on the scale of the surface roughness of the material, the three groups of surface roughness can be distinguished: nanoroughness (<100 nm), microroughness (100 nm–100 μm),

and macroroughness (>100 μm). Each range of roughness may affect the osteointegration processes [44]. Additive manufacturing enables the production of individual bone implants with desired roughness values based on the demand of each patient.

Unfortunately, increased roughness also promotes the adhesion of pathogens. The risk of infection associated with implantation (IAI) depends on the anatomic site, time, and depth of biomaterial application. In the case of completely internal implants (scaffolds) the possible routes of infection are associated with contamination of the implant surface before or during the surgery and hematogenous dissemination from a distant infected site. Bacterial biofilm formed on the implant surface affects osseointegration and poses a risk of systemic infections and septicemia, which can cause patient death. Unfortunately in most cases, the bacterial biofilm is resistant to systemic antibiotic treatment [47,48]. Therefore, it is necessary to produce a multifunctional layer that exhibits a variety of effects: no toxicity to bone tissue (biocompatibility) and toxicity to a broad spectrum of pathogens.

The aim of this research was to deposit a tight and continuous DLC coating on the 316L SS substrate produced in additively manufactured binder jetting. The substrate was designed as a stochastic porous structure based on Voronoi tessellation. Due to the complex structure of the substrate, Radio Frequency Plasma Activated Chemical Vapor Deposition (RF PACVD) technology was used for the deposition of the coating.

The deposition of DLC coatings to increase the biocompatibility of 316L SS substrates is widely described in the world literature [49–55]. The innovativeness of the solutions presented in this paper concerns the method of producing modified substrates (binder jetting technology) and their macrostructure (irregular porous structure). An additional challenge is the high surface roughness that is characteristic of the additive manufacturing process. This topic is important due to the growing interest in the use of additively manufactured porous structures for biomedical applications. Therefore, it is necessary to conduct comprehensive and interdisciplinary research in order to confirm the usefulness of the proposed solutions.

## 2. Materials and Methods

### 2.1. Samples Design

The mathematical definition of the Voronoi tessellation is given by the following equation [16,41,42]:

$$V(p_i) = \{p / d(p,p_i) \leq d(p,p_j), j \neq I, j = 1, \ldots n\}, \tag{1}$$

where:

- $p = \{p_i, \ldots, p\}$ is a set of distinct seed points located in the d-dimensional Euclidean space $R_d$,
- $d(p,p_i)$ represents the Euclidean distance between the location p and seed $p_i$,
- $V(p_i)$ represents the ordinary Voronoi polygon associated with seed pi.

The process of generating a two-dimensional Voronoi tessellation diagram is shown in Figure 1a. Seeds points are randomly generated in the design space. From each point, circles diverge at the same speed and grow outward until neighboring circles impinge on one another. Finally, the Voronoi diagram bounded by line is formed. In 3D space (Figure 1b), each seed grows a polygon or polyhedron, and the diagram is connected by planes. The next step is to determine the diameter of the strun.

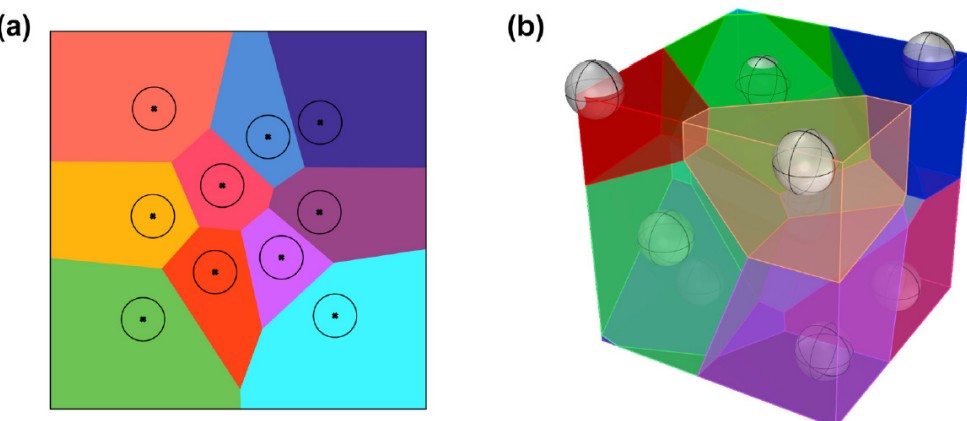

**Figure 1.** Schematic illustration of Voronoi-tessellation: (**a**) 2D diagram; (**b**) 3D diagram [41].

In this study, nTopology (nTopology, New York, NY, USA) software [56] was utilized to design the samples with geometrical dimensions shown in Figure 2. The sample was based on a cylinder (grey) with 10 mm in diameter and 2 mm in height. The design space for the Voronoi structure was a cylinder (green) with 9.7 mm in diameter and 1 mm in height. The diameter of the design space has been reduced by the designed strun diameter so that the Voronoi structure does not exceed the base of the sample. The total height of the sample was 3 mm. The seed points were randomly generated with a predeterminate average distance of 1 mm. The design strun diameter was 0.3 mm.

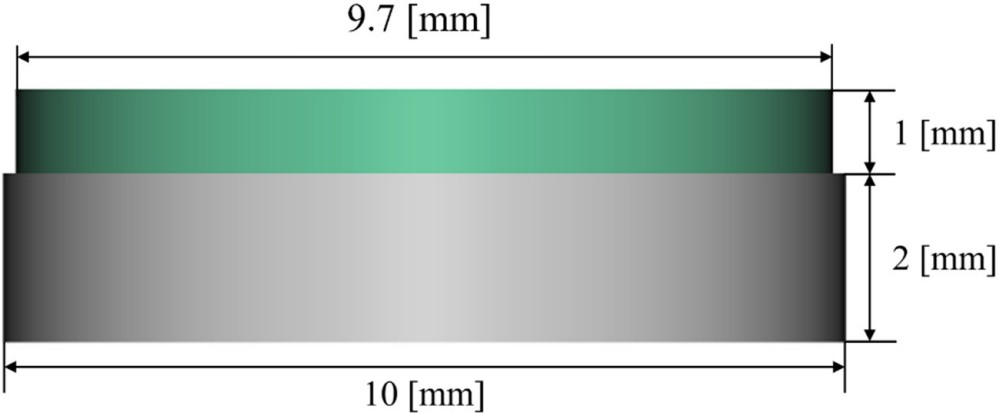

**Figure 2.** Geometrical dimension of samples (back view): grey cylinder- base of the samples, green cylinder- design space.

### 2.2. Powder Characterization

The samples were made from 316L stainless steel powder with an average particle size of 22 μm [57], intended for additive manufacturing in the binder jetting technology (Sandvik Osprey Ltd., Neath, UK). The chemical composition of the powder is shown in Table 1. The scanning electron microscope image (Figure 3A), obtained using PHENOM PRO (Thermo Fisher Inc., Waltham, MA, USA), shows powder morphology. The powder particle size (Figure 3B) was measured using ANALYSETTE 22 MicroTec Plus (Fritsch GmbH, Amberg, Germany) according to PN-ISO 9276-1.

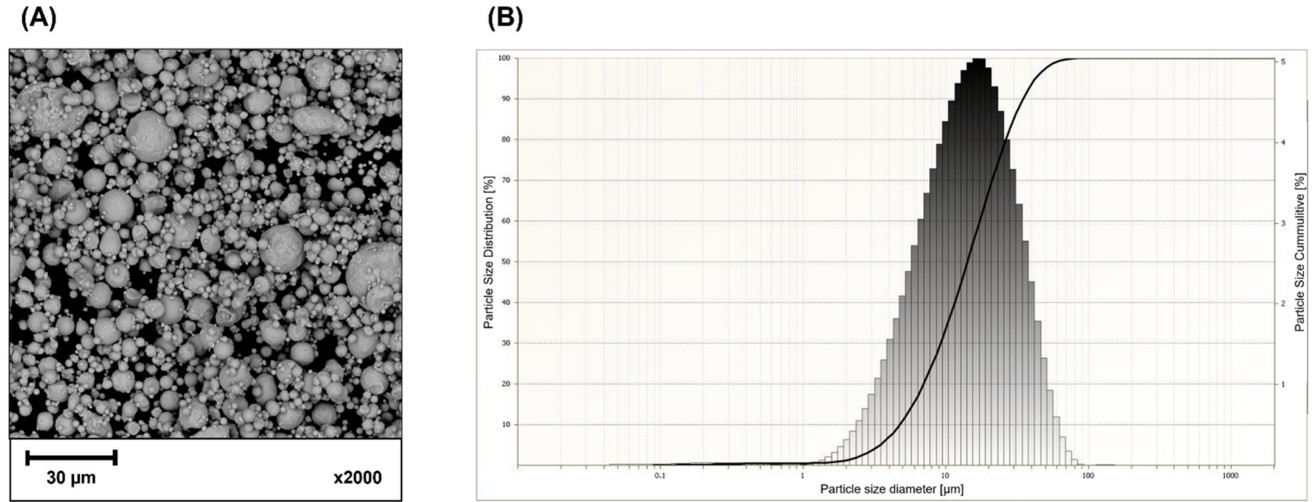

**Figure 3.** Osprey® 316L powder characterization: (**A**) SEM morphology, (**B**) particle size distribution analysis.

**Table 1.** Chemical composition of Osprey® 316L powder [57].

| Element | C | Si | Mn | P | S | Cr | Ni | Mo | Fe |
|---|---|---|---|---|---|---|---|---|---|
| **Weight percent [%]** | ≤0.03 | ≤1 | ≤2 | ≤0.045 | ≤0.03 | 16–18 | 10–14 | 2–3 | Balance |

### 2.3. Fabrication Process

The samples were fabricated using an ExOne Innovent+ (ExOne, Huntington, PA, USA) binder jetting machine. The main parameters of the printing process such as layer thickness, saturation, recoat speed, roller speed, and ultrasonic intensity are shown in Table 2.

**Table 2.** Manufacturing parameters of the binder jetting process.

| Layer Thickness | Saturation | Recoat Speed | Roller Speed | Ultrasonic Intensity |
|---|---|---|---|---|
| 40 μm | 70% | 50 mm/s | 500 rpm | 100% |

The job box was heated at 200 °C for 8 h in a drying oven DX412C (Yamato Scientific Co., Ltd., Tokyo, Japan). The excess powder was removed with compressed air. Nabertherm environment-controlled furnace (Nabertherm GmbH, Lilienthal, Germany) was used to sinter the printed samples after depowderization. The process was carried out in an argon environment, which can cause the formation of nitrides on the surface [58]. Argon gas was pumped into the furnace chamber throughout the sintering cycle until the total temperature decreased. The sintering temperature profile is shown in Figure 4.

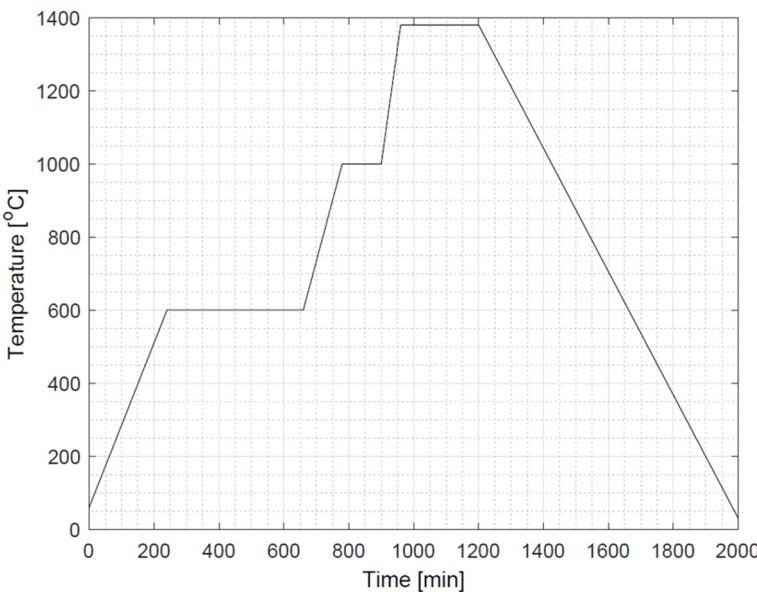

**Figure 4.** Applied temperature profile for sintering of 316L SS green parts printed by binder jetting.

### 2.4. Diamond-Like Carbon Coating Deposition

Before the modification, the samples were ultrasonically washed in isopropyl alcohol for 10 min and then dried in compressed air.

To better clean the surface and raise its temperature, a plasma pre-treatment in RF PACVD equipment (TUL, Lodz, Poland) was used. The substrates were placed directly on the RF electrode. The treatment was carried out under the pressure of 2 Pa, using the 1000 V negative bias voltage of the RF electrode. The treatment time was 10 min.

The DLC coating was deposited using the same RF PACVD equipment [29,30]. The chamber pressure was 40 Pa with a methane gas flow rate of 60 sccm. The negative bias voltage was 700 V. The deposition time was 5 min. Basic modification parameters are presented in Table 3.

**Table 3.** Basic parameters of pre-treatment and DLC coating deposition process.

|  | Pre-Treatment | Deposition of DLC Coating |
|---|---|---|
| **Pressure** | 2 Pa | 40 Pa |
| **RF generator** | 500 W | 1000 W |
| **Negative bias voltage** | 1000 V | 700 V |
| **Deposition time** | 10 min | 5 min |
| **Type of atmosphere** | - | $CH_4$ |

### 2.5. Optical Microscopy

The dimensional analysis of Voronoi structures after the sintering stage was performed using a NIKON MA200 (Nikon, Minato, Tokyo, Japan) optical microscopy. The measurement was performed in three random places. The strun diameter and the pore size are assessed.

### 2.6. SEM Microscopy

The surface morphology of the structure before and after modification of DLC coating was assessed using scanning electron microscopes PHENOM PRO (Thermo Fisher Inc., Waltham, MA, USA) and JEOL JSM-6610LV (JEOL Ltd.; Musashino, Akishima, Tokyo, Japan).

SEM microscopy was also used to analyze the porosity of the internal structure after the sintering process. The sample without DLC coating was cut using a diamond blade cutter with a water bath to prevent heating of the sample. The obtained cross-sections

were fixed in a black phenolic fiberglass resin (PRESI, Eybens, France). Next, the cross-sections were ground using a handheld wet grinder and abrasive papers with grain sizes in the range of P240 to P1200. The last stage was polishing with the use of diamond paste with a grain size of 3 μm (Metalogist, Warsaw, Poland). After polishing, the cross-section was washed with isopropyl alcohol and dried with compressive air. The cross-sections were examined using a scanning electron microscope PHENOM PRO (Thermo Fisher Inc., Waltham, MA, USA).

SEM microscopy was also used to analyze the thickness and continuity of the DLC coating. The sample with DLC coating was cut with a precision saw with water-cooled $Al_2O_3$ cutting disc. The samples were fixed into the conductive resin Polyfast (Struers, Copenhagen, Denmark) and subsequently ground and polished. The final step of polishing was mechanical-chemical polishing using colloidal silica OP-U (Struers, Copenhagen, Denmark). After polishing, the cross-section was washed with isopropyl alcohol and dried with compressive air. Samples were examined using a scanning electron microscope ZEISS Ultra Plus (ZEISS, Oberkochen, Germany) equipped with an energy-dispersive spectrometer Oxford X-Max20 (Oxford Instruments, Abingdon, UK).

PHENOM PRO has a magnification in the range of 160–350,000× and a resolution of ≤6 nm. JEOL JSM-6610LV has a magnification in the range of 5–300,000× and a resolution of ≤4 nm. ZEISS Ultra Plus has a magnification in the range of 12–1,000,000× and a resolution of 1 nm. The value of resolution in SEM microscopy is influenced by many factors related to the measurement, e.g., accelerating voltage or various types of electromagnetic disturbances.

### 2.7. Confocal Microscopy

The substrate surface topography before and after modification of DLC coating was examined using a confocal microscope LEXT OLS4000 (Olympus, Shinjuku-ku, Tokyo, Japan). The measurement was carried out for three random strun of the Voronoi structure. For each strun, the roughness profiles along and across the structure are plotted. Due to the small size of the strun, the measuring distance was shorter than that provided for in the ISO 4287 standard [59]. The TalyMap Platinium (Taylor Hobson, Leicester, UK) software was used to analyze the data. LEXT OLS400 assures accuracy of 0.2 + L/100 μm (where L is the measuring length in μm).

### 2.8. Raman Spectroscopy

Raman studies were conducted on the InVia Raman microscope (Renishaw, Gloucestershire, UK). The spectra were analyzed in rage 900 to 2000 $cm^{-1}$ with an excitation wavelength of 532 nm and an exposure time from 10 to 30 s. Each spectrum was deconvoluted into two peaks (D and G) by using spectral analysis software (Peak Fit v4.11) to calculate the intensity ratio ID/IG.

### 2.9. HRTEM Microscopy

High-Resolution Transmission Electron Microscopy was used to evaluate the physicochemical characterization of the DLC coating. Tests were performed using a scanning transmission electron microscope TECNSI $G^2$ F20 (FEI Company, Hillsboro, OR, USA) in the accredited laboratory of the Institute of Metallurgy and Material Engineering Polish Academy of Science in Krakow. The bright field observation was carried out (according to the P/19/IB-05 procedure) for a cross-section coating cut from the spatial structure using the focused ion beam (FIB). The electron diffractions from larger crystallites were performed (in accordance with the procedure P/19/IB-06) and maps of the distribution of the components were made.

## 3. Results and Discussion

### 3.1. Dimensional Analysis

Figure 5 shows the macrophotography of the fabricated Voronoi structure in comparison with the CAD model. Compare to the design model, the interconnected complex spatial network of the Voronoi structure was accurately duplicated by the binder jetting technology.

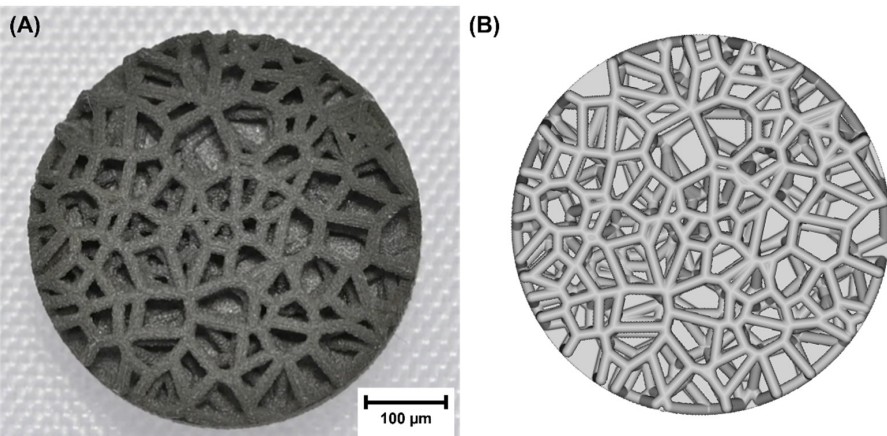

**Figure 5.** Macrophotography of the fabricated samples with Voronoi structure (**A**) in comparison with CAD model (**B**).

The dimensional analysis of the fabricated Voronoi structure was performed using optical microscopy. A characteristic aspect of the binder jetting technology is the shrinkage of the sintered element. Therefore, the element is printed with a certain magnification in relation to the CAD project. In the case in question, a magnification of 20% was used.

Optical images show that after the sintering process, the dimensions of the struns (Figure 6) and pores size (Figure 7) of the Voronoi structure were larger than the design. As shown in the literature and our experience, the shrinkage can be different on each axis (non-linear character of shrinkage) [20,22–25]. In the case of porous structures, it is more difficult to predict the shrinkage scale than in the case of dense components. A larger strun diameter increases the relative density of the structure and thus reduces its porosity. Increased strun diameter can also affect the shape of the pores and connections between the struns. All these aspects introduce additional irregularity in the macrostructure of porous structure which can be added advantage for biomedical applications. However, the dimensional-shape accuracy of the samples was not a necessary condition for the next stage of the research.

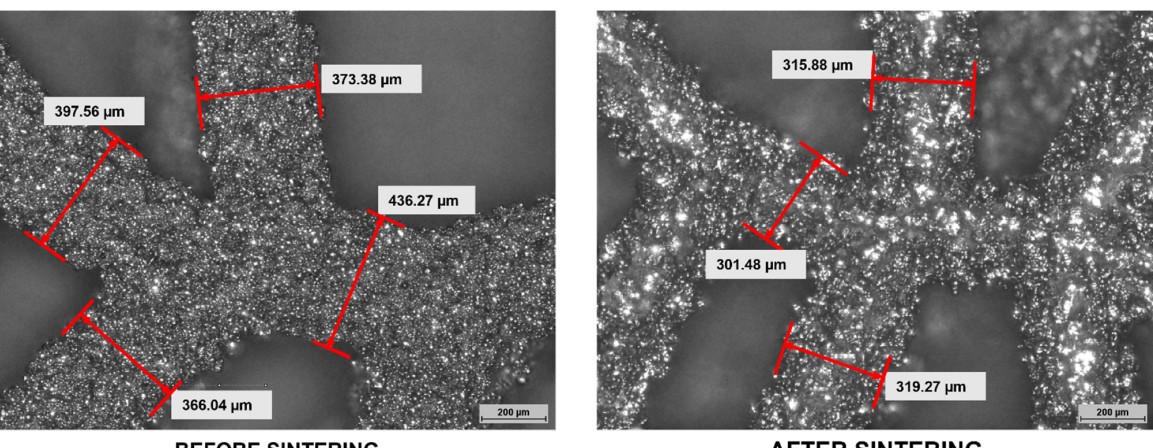

**Figure 6.** The dimensions of the struns of the Voronoi structure obtain from optical microscopy.

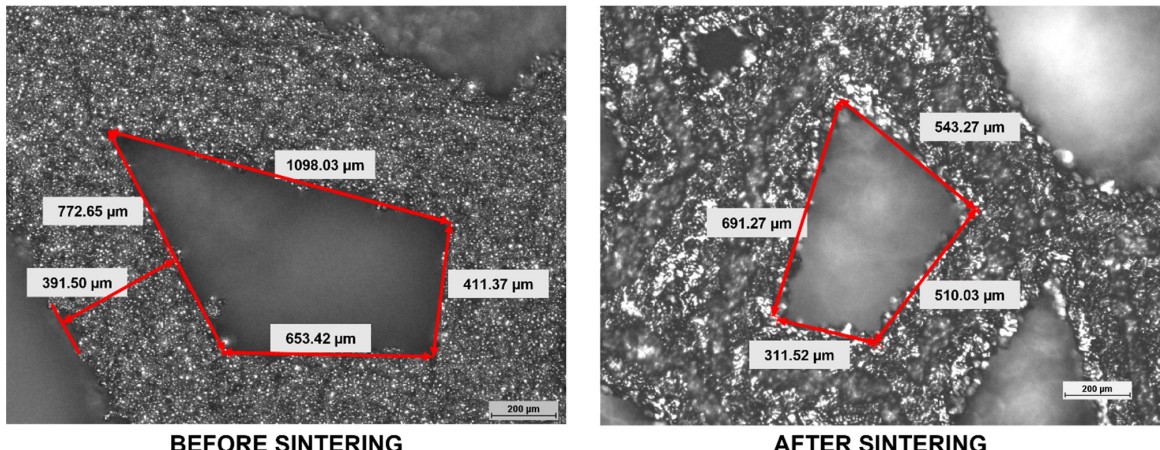

**Figure 7.** The dimensions of the pores of the Voronoi structure obtain from optical microscopy.

*3.2. Surface Morphology*

A scanning electron microscope was used to characterize the surface morphology and local features of the substrates before (Figure 8A–C) and after (Figure 8D–F) modification with DLC coating. Figure 8F was created as a combination of several SEM images obtained for different levels of the analyzed structure.

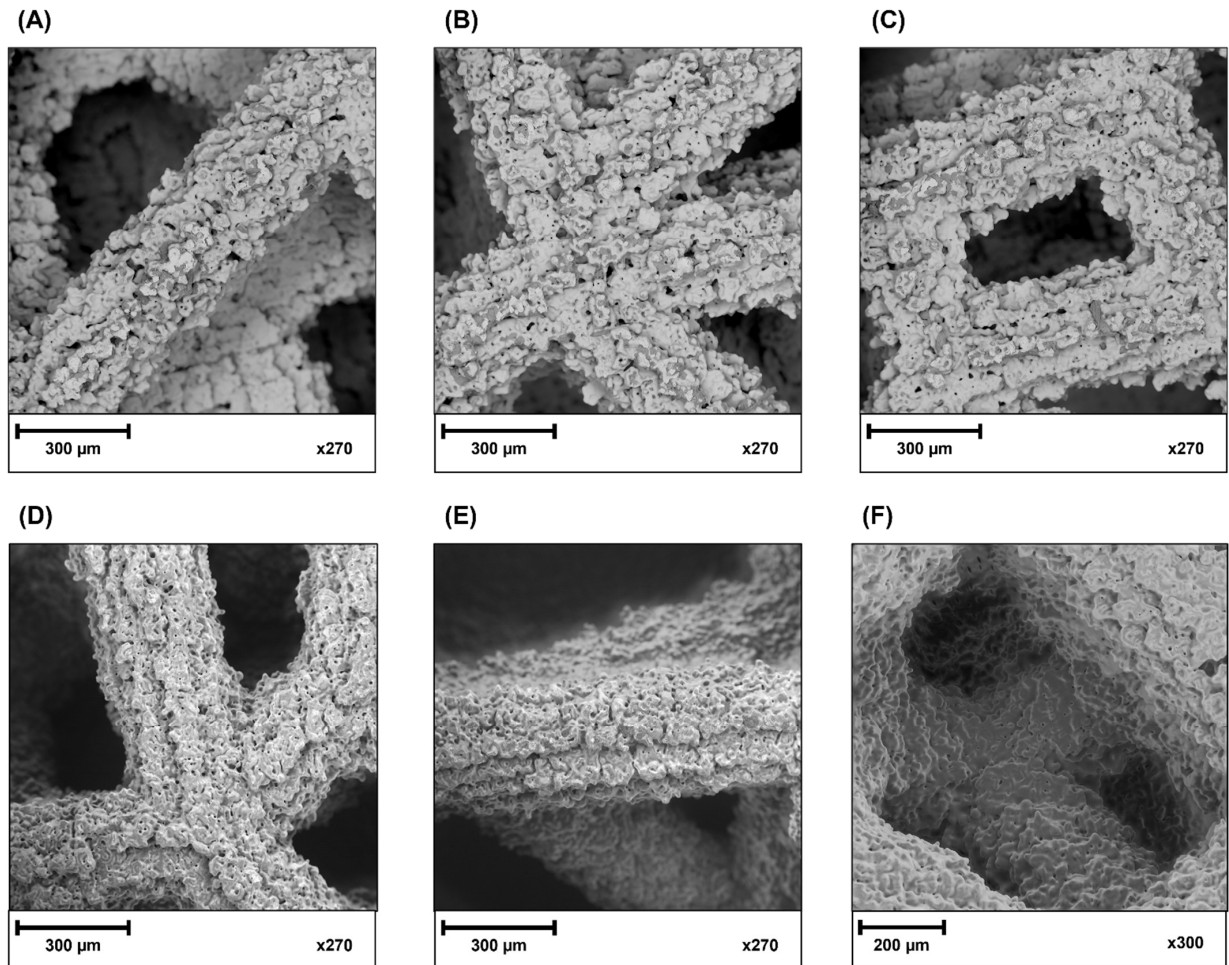

**Figure 8.** The SEM (PHENOM PRO) images of the Voronoi structure: (**A–C**) before and (**D–F**) after modification with DLC coating. Figure 8F was created as a combination of several SEM images obtained for different levels of the analyzed structure.

The 316L SS powder used for sample fabrication has aspherical grains morphology with numerous satellites as shown in Figure 3A. Small particles (<20 μm) with irregular morphology show higher interparticle adhesion caused by mechanical interlocking of angular particles, which leads to the formation of agglomerates. This fact affects of the uniformity of the powder bed density and the penetration of the binder. On the other hand, small powder particles initiate sintering faster than larger particles (>50 μm) and lead to faster densification. All of these have a significant impact on the surface morphology, roughness, and density (internal porosity) of the element after sintering [23,25].

It was observed the rough surface morphology of the struns, was caused by powder grains agglomerates (Figure 8A–C). The powder grains in the same agglomerate sintered to the final stages. Due to the large spacing, the diffusion and necking between adjacent agglomerates are limited, which causes large roughness. The significant observation is the similarity of SEM images of the surface of struns that are located at different levels of the spatial structure (Figure 8D–F). This allows concluded that the DLC coating was successfully deposited on the surface of the Voronoi structure.

Figure 9 shows the SEM images of DLC coatings deposited on the Voronoi structure with different magnifications. It was observed that DLC coating was successfully deposited on the surface of the structure despite its complexity. The image suggests that the coating makes the surface smoother. The SEM images of DLC coatings show that the continuity of the DLC coating is clearly visible. Along with the scale reduction from 200 μm to 10 μm and penetration into the sample, the morphology of the DLC coating shows the result of the plasma washing of the complex shape of the sample, which ensures its continuity. Figure 9C is the best visible effect of the DLC coating, which significantly reduces the porosity of the coated sample. In order to confirm this fact, an analysis of the surface topography (Section 3.4) and an analysis of the cross-sections of the samples after modification with the DLC coating (Section 3.5) was performed.

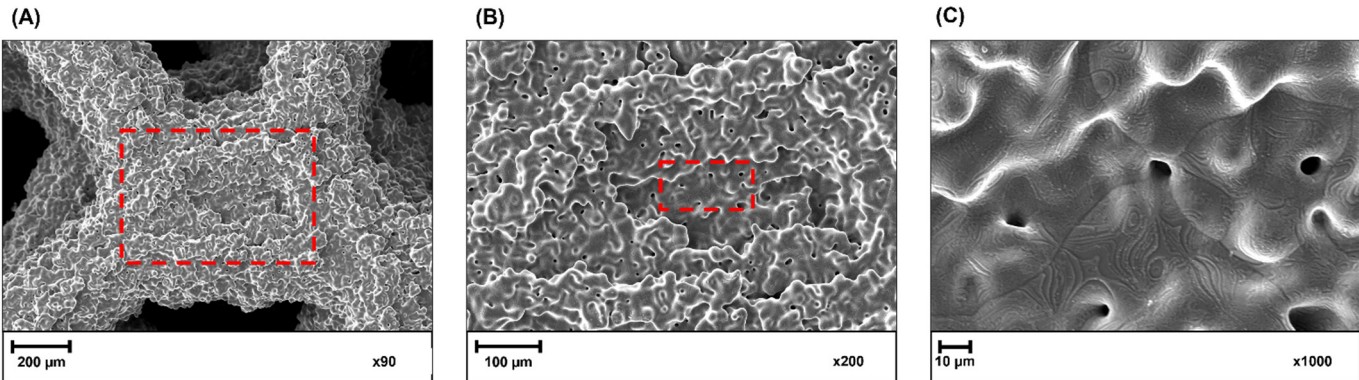

**Figure 9.** The SEM (JEOL JSM-6610LV) images of DLC coatings deposited on Voronoi structure with magnification (**A**) ×90, (**B**) ×200, and (**C**) ×1000.

### 3.3. Internal Porosity

Figure 10 shows the cross-section of the sample without DLC coating and SEM images of the struns. Cross-section analysis shows the complexity of the connections between the struns. The image of a single strun approximates the characteristics of the surfaces not visible in the previous analysis of surface morphology (Section 3.2). On this basis, it can be concluded that struns have a spatial morphology of uniform roughness.

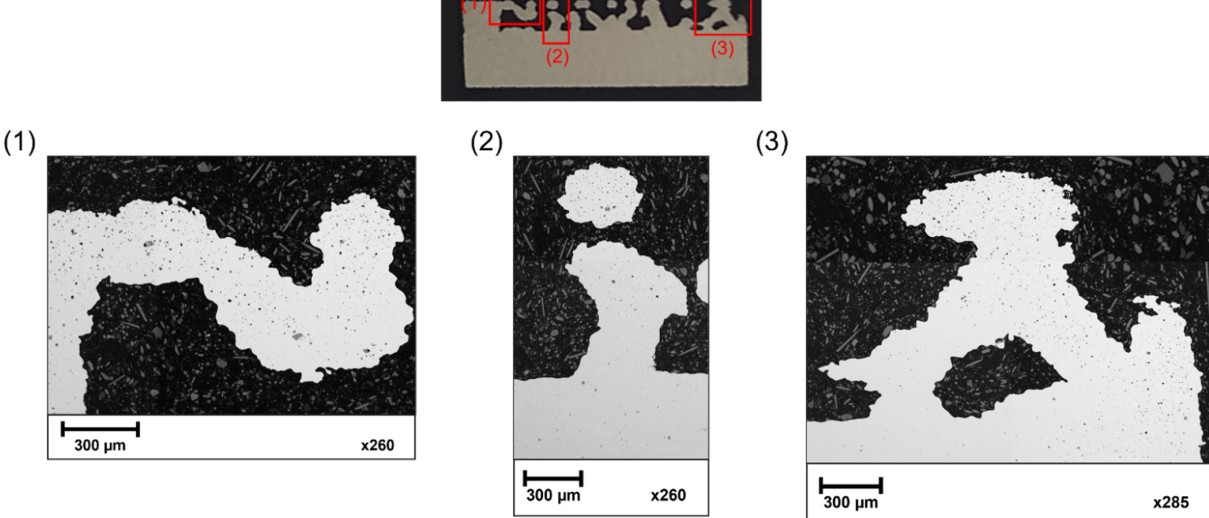

**Figure 10.** The SEM (PHENOM PRO) image of struns of the Voronoi structure before modification with DLC coating was obtained from a cross-section.

In addition, the SEM images of the cross-section were used to analyze the internal porosity of the structure (Figure 11). The main challenge for additive manufacturing technologies is to produce elements with low internal porosity. The research shows, that internal porosity has a significant impact on the mechanical properties of the manufactured elements [20,22,23,25]. There are two types of internal pores: spherical (commonly due to trapped gas or material evaporation) and irregular (due to shrinkage, lack of fusion, or material feed shortage) [22]. Tillman et al. [37] also pay attention to the so-called residual porosity- open pores on the surface of binder jetting elements. They showed that the pore size of the surface has a significant influence on the growth mechanism of DLC coating.

It was observed that most of the inner pores are located in the upper layers of the struns. Inner pores were also located at the periphery of the cylindrical base of the sample. Most of the pores have a spherical morphology. The small number of internal pores in the cross-section may indicate a good packing density of the powder bed at the printing stage and the appropriate selection of the sinter parameters [23,24]. Moreover, no open pores were found in the near-surface area that could affect the coating depositing process, as described by Tillman et al. [37].

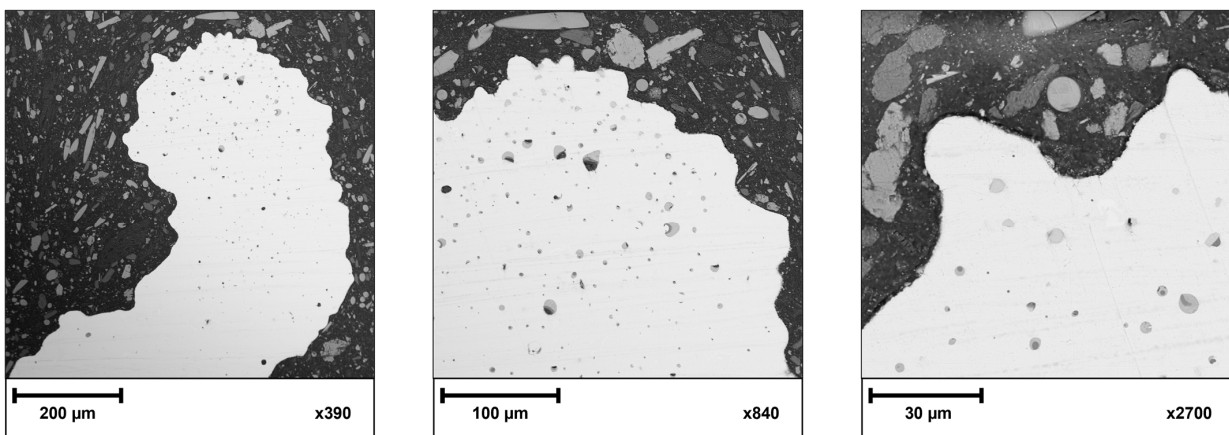

**Figure 11.** The SEM (PHENOM PRO) images of the internal porosity of the strun of the Voronoi structure.

### 3.4. Surface Topography

A confocal microscope was used to analyze the surface topography of the substrates before and after modification with DLC coatings. Figures 12 and 13 show surface topography images obtain for one of the strun of Voronoi structure before and after modification with DLC coating (respectively) and roughness profiles obtain for four different places of the strun. Table 4 presents the roughness parameters obtained based on the roughness profiles for three struns according to ISO 4287 [59].

Research shows that increased roughness of surfaces has an important role in osteointegration processes [44–46]. Additive manufacturing enables the production of individual bone implants with desired roughness values. Based on the data from Table 4 it was observed that the average roughness of the surface of the strun before modification with DLC coating was higher in comparison with the strun after modification. Nevertheless, values of Ra and Rz were in the microroughness range and are typical for additive manufacturing structures [20–25]. It suggests that the DLC coating partially smoothed the structure by filling the space between adjacent agglomerates.

**Table 4.** The roughness parameters obtained based on the roughness profiles for struns before and after modification with DLC coatings.

| | S1 | S2 | S3 | AVRAGE [μm] | S1_DLC | S2_DLC | S3_DLC | AVRAGE [μm] |
|---|---|---|---|---|---|---|---|---|
| | | | | ALONG ROUGHNESS PROFILE | | | | |
| **Ra [μm]** | 7.18 | 8.61 | 9.07 | **8.29** | 3.10 | 5.40 | 1.58 | **3.36** |
| **Rz [μm]** | 44.10 | 37.80 | 69.80 | **50.57** | 15.00 | 25.00 | 16.90 | **18.97** |
| | | | | ACROSS ROUGHNESS PROFILE | | | | |
| **Ra [μm]** | 3.54 | 7.65 | 6.28 | **6.43** | 4.45 | 2.96 | 4.39 | **4.04** |
| | 6.17 | 6.84 | 6.69 | | 4.08 | 3.68 | 5.53 | |
| | 8.18 | 8.50 | 3.99 | | 3.59 | 2.97 | 4.68 | |
| **Rz [μm]** | 18.70 | 41.70 | 34.20 | **32.76** | 20.90 | 14.90 | 16.20 | **18.89** |
| | 28.50 | 32.10 | 27.40 | | 20.90 | 18.30 | 24.90 | |
| | 53.60 | 36.60 | 22.00 | | 16.20 | 13.10 | 24.60 | |

Where: S—a symbol of the strun before modification; S_DLC—a symbol of the strun after modification.

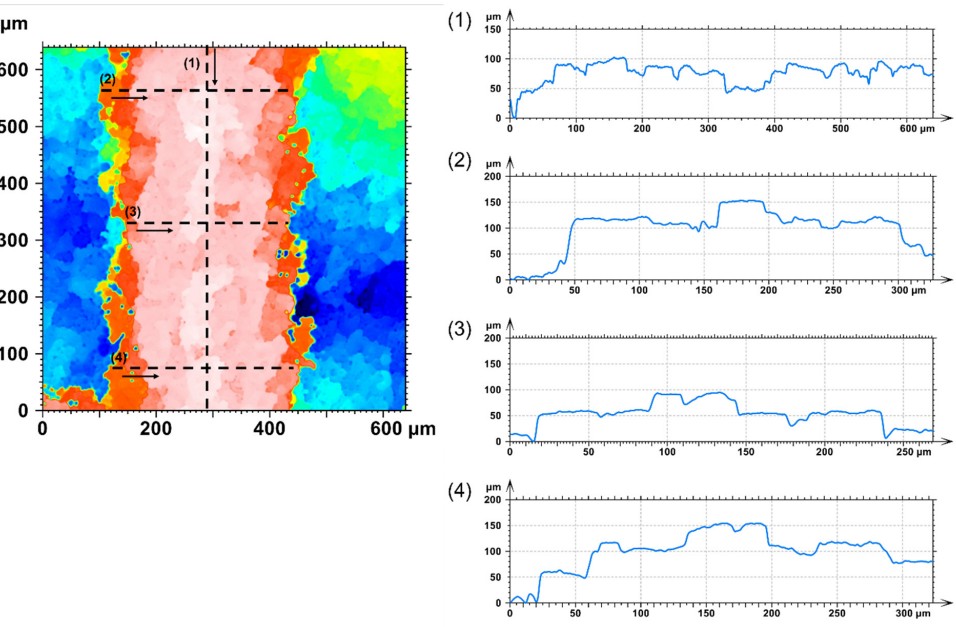

**Figure 12.** The surface topography image obtained for one of the strun of Voronoi structure before modification with DLC coating and roughness profiles (black arrows indicate the direction in which to draw the profile).

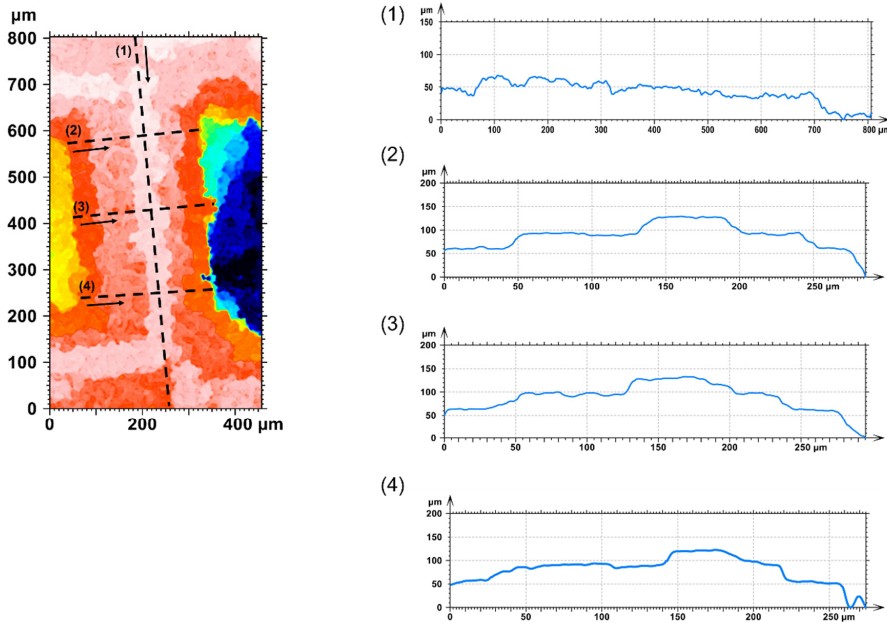

**Figure 13.** The surface topography image was obtained for one of the strun of the Voronoi structure after modification with DLC coating and roughness profiles (black arrows indicate the direction in which to draw the profile).

### 3.5. Thickness and Continuity of the DLC Coating

Figure 14 shows SEM images of the cross-section of the struns with DLC coating. Cross-section analysis indicates a continuous structure of the DLC coating of varying thickness. Based on image analysis, it was found that the layer thickness ranged from 30 nm to 230 nm. The thickness of the coating depends on the radius of curvature of the surface. For a smaller radius of surface curvature, the deposited coating is thinner. However, the SEM image shows the continuity of the deposited coating and confirms that the coating smoothes the surface roughness.

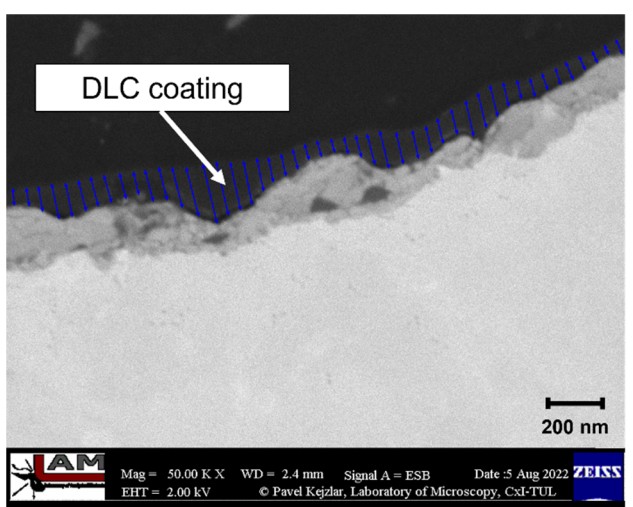
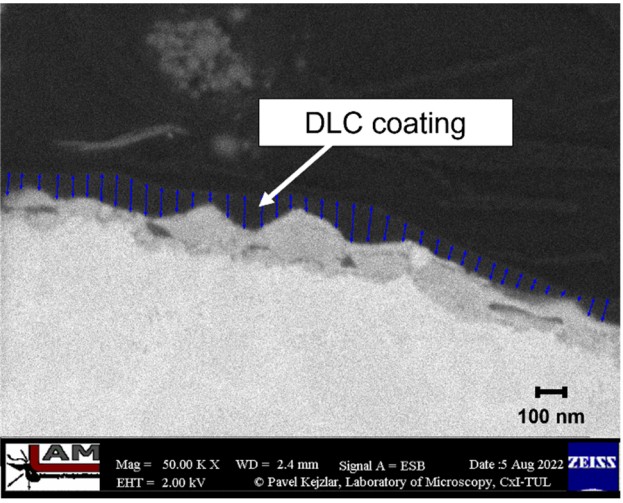

**Figure 14.** SEM image (ZEISS Ultra Plus) of DLC coating obtained from a cross-section.

### 3.6. The Chemical Structure of DLC Coating

For the chemical structure analysis of the DLC coating, Raman spectroscopy was used. Figure 15 shows the results of the examined DLC coating. The obtained Raman spectra were deconvolved into four peaks: D1 (1180 cm$^{-1}$), D2 (1350 cm$^{-1}$), G1 (1532 cm$^{-1}$), and G2 (1590 cm$^{-1}$), which is reflected in the literature [30]. The result of Raman spectroscopy

shows the presence of a typical DLC coating on the surface of the tested metal sample described in the literature [60–62]. The ID/IG = (ID1 + ID2)/(IG1 + IG2) ratio equal to 1.08 additionally confirms the typical chemical structure of DLC coatings which affects its physicochemical and material properties, increases its biocompatibility, and from the diffusion barrier that protects against metallosis [26,35,63–65].

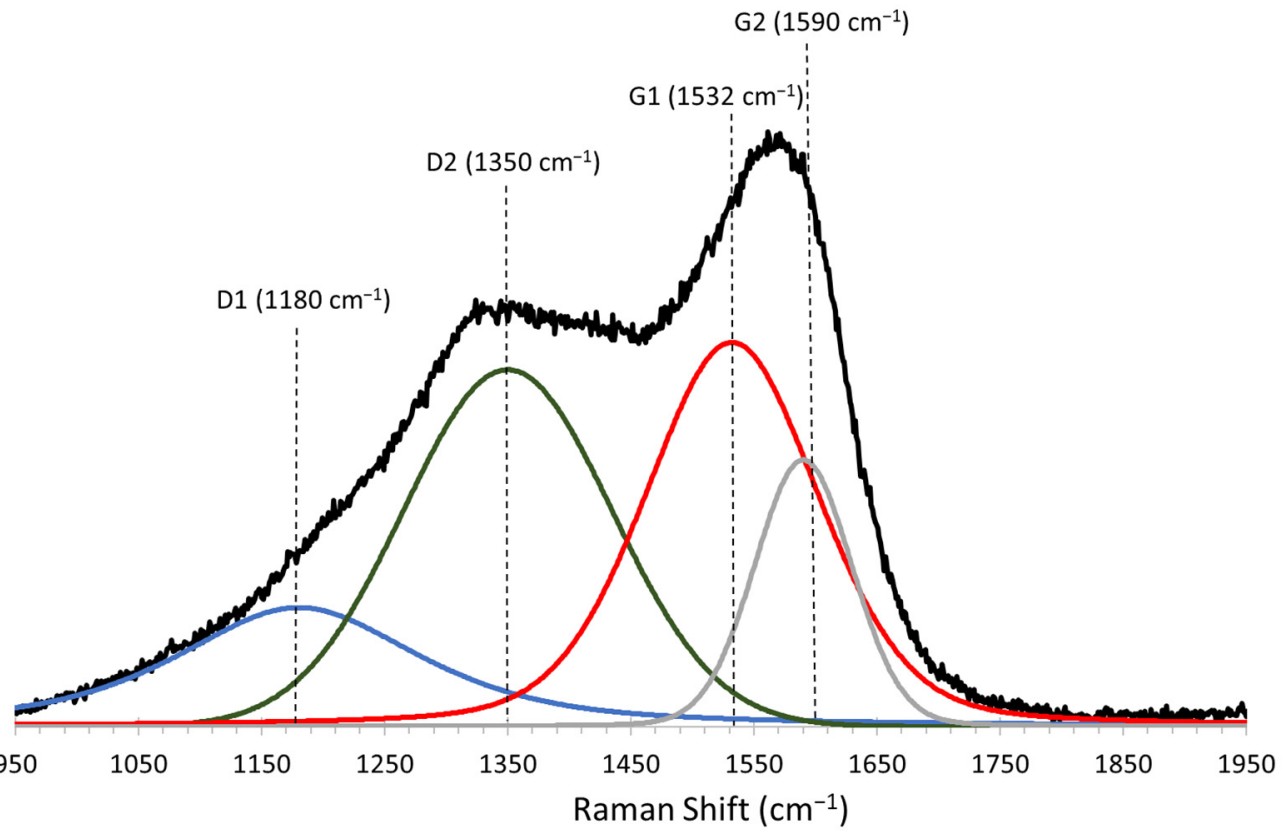

**Figure 15.** Raman spectrum of DLC coating manufactured by RF PACVD method- IG/IG = 1.08.

### 3.7. Microstructure and Chemical Composition

High-Resolution Transmission Electron Microscopy was used to evaluate the physicochemical characterization of the DLC coating. Figure 16A shows the TEM/BF microstructure image, which presents an overview of the microstructure of the near-surface area of binder jetting 316L SS strun with DLC coating. Figure 16B shows the STEM/HAADF image with corresponding maps presenting the distribution of C, O, Ni, Fe, Cr and S (Figure 16C–H) [60–69].

The results indicate a continuous image of the DLC coating with content chromium oxide and silicon dioxide. The thickness of examined DLC is about 200 nm but in comparison with SEM images of the cross-section analysis (Figure 14), we observed the changes in the thickness (from 30 nm to 230 nm) of the coating depending on its curvature.

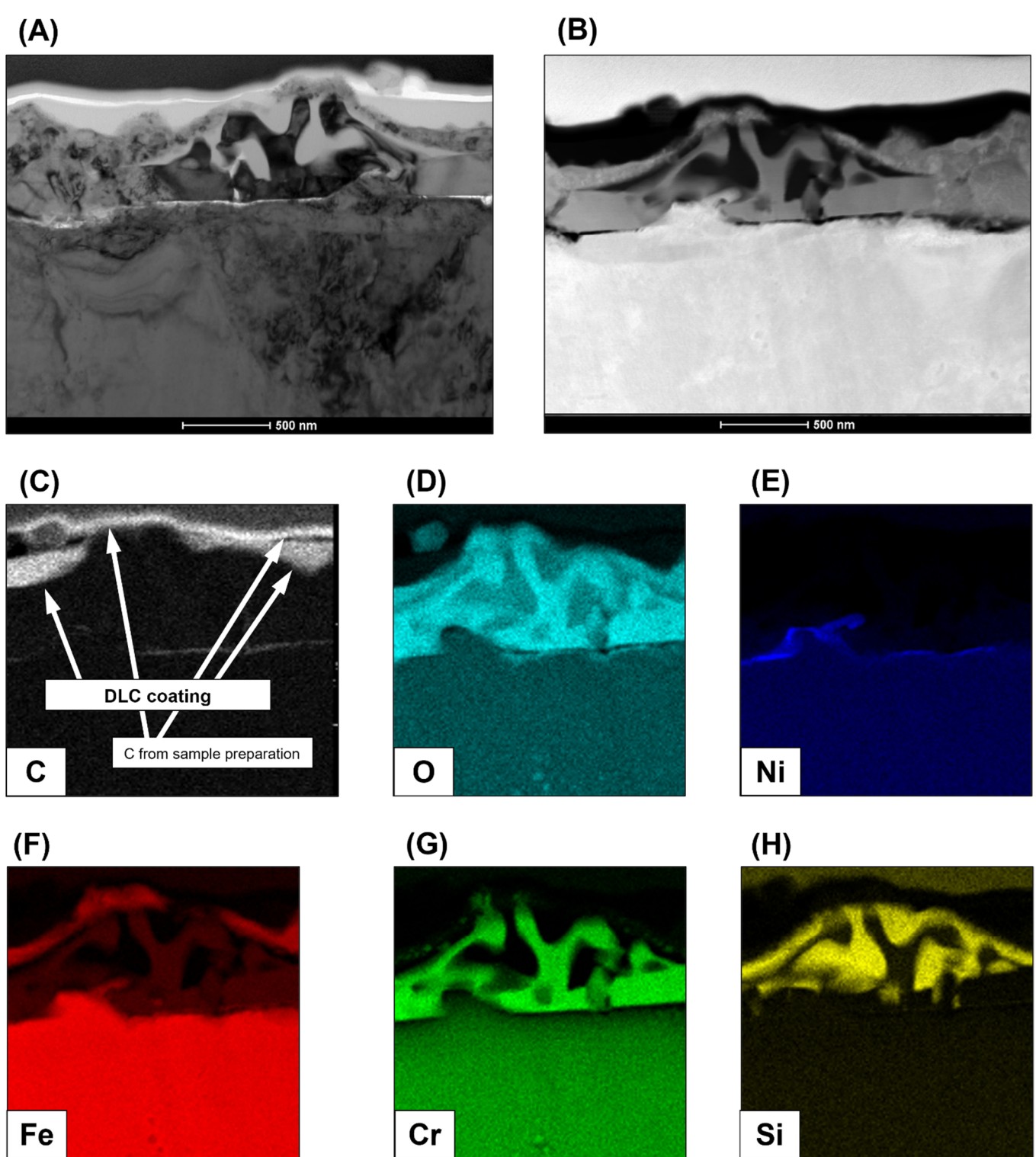

**Figure 16.** TEM/BF (**A**) and STEM/HAADF (**B**) images of the DLC coating and corresponding maps presenting the distribution of C (**C**), O (**D**), Ni (**E**), Fe (**F**), Cr (**G**) and S (**H**).

## 4. Conclusions

The tested spatial structure was designed to increase the osseointegration potential of the surface of medical implants. The spatial arrangement of the Voronoi structure is intended to mimic the bone matrix. An additional advantage is the surface microroughness obtained as a result of fabrication using the binder jetting technology. Research shows that high surface roughness promotes the adhesion of cells and proteins. Unfortunately, it

also promotes the adhesion of pathogens. Bacterial biofilm formed on the implant surface affects osseointegration and represents a risk of systemic infections and septicemia, which can cause patient death. Another risk associated with metal implants is biocorrosion and ion emission which lead to metallosis and allergic reactions. According to this, the conduct of comprehensive and interdisciplinary research on the surface modification of additively manufactured porous structures for biomedical applications is very important.

In this study stochastic porous structures based on Voronoi tessellation were made using binder jetting technology from 316L SS powder and modified using DLC coating. The evaluation of binder jetted 316L SS substrate and deposited DLC coating allows for the following conclusions:

- Compared to the design model, the interconnected complex spatial lattice of the Voronoi structure was accurately duplicated by the binder jetting technology;
- The dimensions of the struns and pores size of the Voronoi structure were larger than the design, which reduces its porosity, and affects the shape of the pores and connections between the struns;
- The specificity of the spatial structure design, the material used, and the manufacturing technology influenced the morphology and topography of the surface. In SEM analysis, it was observed that the rough surface morphology of the struns, was caused by powder grain agglomerates. The powder grains in the same agglomerate sintered to the final stages, but due to the large spacing, the diffusion and necking between adjacent agglomerates were limited;
- SEM images of DLC coatings obtained for different levels of the analyzed structure show that the DLC coating was successfully deposited on the surface of the Voronoi structure;
- The main challenge for additive manufacturing technologies is to produce elements with low initial porosity. The inner pores were located in the upper layers of the struns and at the periphery of the cylindrical base of the sample. In the near-surface area, no open pores were observed that could affect the coating deposition process. Most of the pores have a spherical morphology. The small number of internal pores in the cross-section may indicate a good packing density of the powder bed at the printing stage and the appropriate selection of the sinter parameters;
- Based on the data from the confocal microscopy analysis, it was observed that the mean roughness of the surface of the strun before modification with DLC coating was higher in comparison with the strun after modification. The DLC layer caused a partial smoothing of the structure by filling the space between adjacent agglomerates. This was highlighted in the cross-sectional analysis of SEM and HR TEM;
- SEM cross-section analysis and HR TEM analysis indicate a continuous DLC coating with thicknesses from 30 nm to 230 nm. It was observed that coating thickness depends on the radius of curvature of the surface;
- The result of Raman spectroscopy shows the presence of a typical DLC coating on the surface of the tested sample with ID/IG = 1.08;
- The HR TEM results indicate a continuous image of the DLC coating with content chromium oxide and silicon dioxide.
- This detailed HR TEM (Figure 16) and SEM images of the cross-section analysis (Figure 14) confirm that the obtained DLC coating is continuous with changing thickness depending on the curvature of the test sample.

The limitations of the presented solutions result from the features characterizing the process of the additive production of binder jetting from metal powders. It has primarily a high surface roughness after the sintering process, limitations in the minimum dimensions of the components of the designed structures, and the possibility of porous spaces at the outer surface. However, the advantages of this method outweigh its disadvantages, so it is important to conduct further research in this area. Future research would be extended by biocorrosion and cytotoxicity studies on selected cell lines. The research should also focus on the use of other geometric, technological, and material solutions.

**Author Contributions:** Conceptualization, B.B. and K.M.; methodology, D.L., B.B., W.K., J.G., L.S., T.S. and K.M.; validation, D.L., B.B. and K.M.; formal analysis, D.L., B.B., W.K., L.S. and K.M.; investigation, D.L., W.K., J.G., L.S. and T.S.; writing—original draft preparation, D.L.; writing— review and editing, B.B. and K.M., visualization, D.L., W.K and L.S.; supervision, B.B. and K.M. All authors have read and agreed to the published version of the manuscript.

**Funding:** This research received no external funding.

**Institutional Review Board Statement:** Not applicable.

**Informed Consent Statement:** Not applicable.

**Data Availability Statement:** Not applicable.

**Conflicts of Interest:** The authors declare no conflict of interest.

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
