# Peer review of "The DLC Coating on 316L Stainless Steel Stochastic Voronoi Tessellation Structures Obtained by Binder Jetting Additive Manufacturing for Potential Biomedical Applications"

_coatings, doi:10.3390/coatings12101373_

Round 1

Reviewer 1 Report

This article presents analyical results of DLC coating deployed on porous structure parts made by binder jetting method. A series of analytical results show that DLC coating can contribute to surface modification of the manufactured part.

Below are some comments for this article:

1. (Line 219-224, Line 344-351) What is the D50 value of the applied powder? Since the powder size distribution would have much influence on the surface roughness of the printed part, maybe a suggested D50 value could be proposed for the binder jetting process.

2. (Line 316-329) Figure 5 is a bit confusing because the structure are not identically similar. If the sample is fabricated using the CAD model, what is the reasons that caused this difference?

In Figure 6 and Figure 7, dimensions before and after sintering process are compared to illustrate the concept of pore shrinkage after sintering. Is the results measured at the same position of the sample? If not, the comparison may lose its meaning, for that the data were measured with different basis. 

3. What is the binder adopted in this process? After the sintering process, would the binder be traped in the pores? For a biomedical application, would it be acceptable if residual binder is left in the fabricated part?

4. It is suggested to unify the range of the scale bars in Figure 12 and Figure 13, so that the readers would have better understanding that DLC coating contributes to the smoothen of coated surface.

5. Please correct the spelling error in Line 447 and Figure 16 (DCL -> DLC)

6. Is the word "strun / struns" often used in other reaearches? An alternative expression of "branch / branches" may be considered.

Author Response

Dear Reviewer,
Thank you very much for your comments and suggestion. We are enclosing the answers to your questions. According to suggestions we took the extra time to proofread the language again.

Best regards,

Authors

Reviewer 2 Report

There are a few point which should be clarified prior to publication: 

Line 441: the greater the curvature? Or, the larger the radius of the curvature?

Fig. 14, caption: apart from the hatched area, what are the white, light gray, and dark areas?

Line 449: The obtained ?? according to the literature?

Fig. 16: where do I see the “crystallite diffraction”?

Conclusions (and line 514), check sentence: indicate a “continuous image” ?? of the DLC coating with different thicknesses “in varied” ?? from 30 nm to 230 nm.

Conclusions: amorphous “carbon” (or DLC??) coating

Author Response

(The authors gave the same response as above.)

Reviewer 3 Report

1.      The article title's uppercase and lowercase should be changed according to MDPI format.

2.      The abstract section should be enhanced to include quantitative data.

3.      Please add the abstract's "take-home" message, the current form was insufficient.

4.      Rearrange the keywords so that they are in alphabetical order.

5.      Make the each of keywords with lowercase font following MDPI format, revise it.

6.      Abbreviation as a keyword is not recommended and encouraged to be changed become a stand for its abbreviation.

7.      What is the novel of the present study? It works have been widely studied in the past. Nothing something really new in the present form related to DLC coating on stainless steel. The lack of novel seems to make the present study like to replication/modified study. The authors need to detail their novelty in the introduction section. It is a major concern for rejecting this paper.

8.      The work, novelty, and constraints of relevant previous research must be explained in the introduction section to highlight the research gaps that the present study aims to fill.

9.      Line 50-53 needs reference to support this explanation related to metals materials in implant application. The suggested reverence should be adopted as follows: Ammarullah, M. I.; Afif, I. Y.; Maula, M. I.; Winarni, T. I.; Tauviqirrahman, M.; Jamari, J. Tresca Stress Evaluation of Metal-on-UHMWPE Total Hip Arthroplasty during Peak Loading from Normal Walking Activity. Mater. Today Proc. 2022, 63, S143–6. https://doi.org/10.1016/j.matpr.2022.02.055

10.   Rather than relying just on the predominate text as it already exists, the authors could incorporate more illustrations as figures in the materials and methods section that illustrate the workflow of the current study.

11.   Manufacturer, country, and specification information for experimental setup should be presented with more specificity.

12.   The inaccuracy and tolerance of the experimental equipment used in this inquiry are critical details that must be included in the article.

13.   An evaluation of the findings with similar past research is essential.

14.   What are the limitations of the current work? Please include it before the concluding section.

15.   Compose a paragraph-length conclusion rather than the present form's point-by-point description.

16.   In the conclusion section, discusses future research that is required.

17.   The reference needs to be enriched from the literature published five years back, literature published by MDPI is strongly encouraged.

18.   In the whole of the manuscript, the authors sometimes made a paragraph only consisting of one or two sentences that made the explanation not clearly understood. The authors need to extend their explanation to become a more comprehensive paragraph. In one paragraph, it is recommended to consist of at least 3 sentences with 1 sentence as the main sentence and the other sentences as supporting sentences. See line 98-101.

19.   Because of grammatical faults and linguistic style, the authors must proofread the document. MDPI English editing service would be a solution.

20.   Please ensure that the authors followed the MDPI format correctly; modify the current form and recheck, as well as any other problems that have been highlighted.

Author Response

(The authors gave the same response as above.)

Round 2

Reviewer 3 Report

The revised work is recommended to accept in its present form.